# Evaluation of the Post-Training Hypotensor Effect in Paralympic and Conventional Powerlifting

**DOI:** 10.3390/jfmk6040092

**Published:** 2021-11-01

**Authors:** Felipe J. Aidar, Ângelo de Almeida Paz, Dihogo de Matos Gama, Raphael Fabricio de Souza, Lúcio Marques Vieira Souza, Jymmys Lopes dos Santos, Paulo Francisco Almeida-Neto, Anderson Carlos Marçal, Eduardo Borba Neves, Osvaldo Costa Moreira, Nuno Domingos Garrido, Breno Guilherme Araújo Tinôco Cabral, Filipe Manuel Clemente, Victor Machado Reis, Pantelis Theo Nikolaidis, Beat Knechtle

**Affiliations:** 1Group of Studies and Research of Performance, Sport, Health and Paralympic Sports (GPEPS), Federal University of Sergipe (UFS), São Cristóvão 49100-000, SE, Brazil; fjaidar@gmail.com (F.J.A.); profedf.luciomarkes@gmail.com (L.M.V.S.); jymmys.lopes@gmail.com (J.L.d.S.); acmarcal@yahoo.com.br (A.C.M.); 2Department of Physical Education, Federal University of Sergipe (UFS), São Cristóvão 49100-000, SE, Brazil; raphaelctba20@hotmail.com; 3Program of Physical Education, Federal University of Sergipe (UFS), São Cristóvão 49100-000, SE, Brazil; angelo-paz@uol.com.br; 4Cardiovascular & Physiology of Exercise Laboratory, University of Manitoba, Winnipeg, MB R3T 2N2, Canada; dihogogmc@hotmail.com; 5Physical Education Course, State University of Minas Gerais (UEMG), Divinopolis 37900-106, MG, Brazil; 6Department of Physical Education, Federal University of Rio Grande do Norte (UFRN), Natal 59078-970, RN, Brazil; paulo220911@hotmail.com (P.F.A.-N.); brenotcabral@gmail.com (B.G.A.T.C.); 7Graduate Program of Biomedical Engineering, Federal Technological University of Paraná (PPGEB/UTFPR), Curitiba 80230-901, PR, Brazil; eduardoneves@utfpr.edu.br; 8Institute of Biological Sciences and Health, Federal University of Viçosa, Campus Florestal, Viçosa 35690-000, MG, Brazil; ocostamoreira@gmail.com; 9Research Center in Sports Sciences, Health Sciences and Human Development (CIDESD), Trás os Montes and Alto Douro University, 5001-801 Vila Real, Portugal; ndgarrido@gmail.com (N.D.G.); victormachadoreis@gmail.com (V.M.R.); 10Escola Superior Desporto e Lazer, Instituto Politécnico de Viana do Castelo, Rua Escola Industrial e Comercial de Nun’Álvares, 4900-347 Viana do Castelo, Portugal; filipe.clemente5@gmail.com; 11Instituto de Telecomunicações, Delegação da Covilhã, 1049-001 Lisboa, Portugal; 12School of Health and Caring Sciences, University of West Attica, 12243 Egaleo, Greece; pademil@hotmail.com; 13Institute of Primary Care, University of Zurich, 8091 Zurich, Switzerland; 14Medbase St. Gallen Am Vadianplatz, 9001 St. Gallen, Switzerland

**Keywords:** blood pressure, hemodynamics, hypotension, resistance training

## Abstract

High blood pressure (HBP) has been associated with several complications and causes of death. The objective of the study was to analyze the hemodynamic responses in Paralympic bench press powerlifting (PP) and conventional powerlifting (CP) before and after training and up to 60 minutes (min) after training. Ten PP and 10 CP athletes performed five sets of five repetition maximal bench press exercises, and we evaluated systolic, diastolic, and mean blood pressure (SBP, DBP, and MBP, respectively), heart rate (HR), heart pressure product (HPP), and myocardial oxygen volume (MVO_2_). The SBP increased after training (*p* < 0.001), and there were differences in the post training and 30, 40, and 60 min later (*p* = 0.021), between 10 and 40 min after training (*p* = 0.031, η^2^_p_ = 0.570), and between CP and PP (*p* =0.028, η^2^_p_ = 0.570). In the MBP, there were differences between before and after (*p* = 0.016) and 40 min later (*p* = 0.040, η^2^_p_ = 0.309). In the HR, there was a difference between before and after, and 5 and 10 min later (*p* = 0.002), and between after and 10, 20, 30, 40, 50, and 60 min later (*p* < 0.001, η^2^_p_ = 0.767). In HPP and MVO_2_, there were differences between before and after (*p* = 0.006), and between after and 5, 10, 20, 30, 40, 50, and 60 min later (*p* < 0.001, η^2^_p_ = 0.816). In CP and PP, there is no risk of hemodynamic overload to athletes, considering the results of the HPP, and training promotes a moderate hypotensive effect, with blood pressure adaptation after and 60 min after exercise.

## 1. Introduction

High blood pressure (HBP) has been considered a risk factor related to cardiovascular complications and other diseases, such as sudden death syndrome, stroke, acute myocardial infarction, heart failure, peripheral arterial disease, and chronic kidney disease [1,2,3]. In this regard, blood pressure control with the use of drugs and complementary activities has been widely studied [4,5]. Thus, physical exercise has been used as a form of prevention, control, and nonpharmacological treatment of arterial hypertension, providing an effective and cheaper strategy than pharmacological intervention [4,6]. Therefore, blood pressure reduction below the resting volume after exercise is defined as a hypotensive effect [7,8].

Within the strategies using exercises, studies have indicated that the post-exercise hypotensive effect (PHE) tends to occur regardless of the intensity, and without promoting cardiovascular overload during its practice [9,10]. Among the exercises, resistance exercises have been shown to be effective in controlling and reducing arterial hypertension [2,11,12], providing acute and chronic benefits for hypertensive or nonhypertensive people [6,13]. Among the resistance exercises, we have powerlifting, which is a modality where the one who lifts the most weight wins, and which has presented positive effects on blood pressure levels both for people with no physical disability [11] and for people with disabilities [2].

Powerlifting is characterized as a strength sport where the squat, bench press, and deadlift are performed [14,15], and through the adapted bench press in Paralympic powerlifting (PP) [16]. Studies have shown that powerlifting can be an alternative to prevent and/or control HBP [2,11]; however, to date, there has been no comparison between people with and without disabilities in the powerlifting bench press.

Therefore, our study aimed to analyze the hemodynamic responses generated in PP in comparison with bench press in conventional powerlifting (CP) after training with loads close to five maximum repetitions (5RM) up to 60 min after the end of the training session. In this sense, we raise two hypotheses: (i) powerlifting promotes a hypotensive effect after exercise; (ii) there are hypotensive differences between the conventional and Paralympic powerlifting.

## 2. Materials and Methods

Blood pressure was checked before the intervention (pre-test) for the collections, with a 10 min rest period [2,17,18,19], then the training itself was performed and, after the session, a post-test was done.

Figure 1 exemplifies the experimental design of the study.

### 2.1. Sample

Participants in the study included 10 CP athletes, who trained recreationally, characterized by training at least three times a week but not often participating in official sport competitions, without disabilities, the condition being verified by means of a medical certificate, and 10 PP athletes, all male, participating in the project at the Federal University of Sergipe (UFS) with at least 18 months of training, eligible for the modality and ranked among the top 10 of their categories at the national level, according to the Brazilian Paralympic Committee (CPB) norms. An inclusion criterion to have officially competed at the national level in the last six months was also adopted. Among the eligible deficiencies, four athletes had spinal cord injuries due to accidents with injuries below the eighth thoracic vertebra, two had sequelae due to poliomyelitis, two had malformations in the lower limbs, one was an amputee, and one had cerebral palsy. The characterization of the sample is shown in Table 1.

No fat percentage tests or height tests were performed, given the aforementioned deficiencies, which would make this procedure unfeasible.

Exclusion criteria included the use of some type of illicit ergogenic resource, some type of symptomatic cardiorespiratory or cardiac disease, metabolic changes, or being involved in any process of rapid weight loss before the competition, because this practice can negatively affect physical performance. The athletes participated in the study voluntarily and signed a free and informed consent form, in accordance with the resolution 466/2012 of the National Research Ethics Commission—CONEP, of the National Health Council, following the ethical principles expressed in the Declaration of Helsinki (1964, reformulated in 1975, 1983, 1989, 1996, 2000, 2008, and 2013) of the World Medical Association. This study was approved by the Research Ethics Committee of the Federal University of Sergipe, CAAE: 2,637,882 (date of approval: 7 May 2018).

### 2.2. Instruments

The body mass of the athletes was measured with the subjects in a sitting position, if from the PP Group, or standing, if only from the CP Group, using an appropriate Michetti digital electronic scale, Model Mic Wheelchair (Michetti, São Paulo, SP, Brazil). An official 210 cm long straight bench and 220 cm long 20 Kg bar were used herein (Eleiko Sport AB, Halmstad, Sweden); both pieces of equipment were approved by the International Paralympic Committee (IPC) [21].

To assess muscle strength, the entire procedure was performed in the bench press exercise using an official bench (Eleiko Sport AB, Halmstad, Sweden), approved by the International Paralympic Committee (IPC) [21], and an IPC Olympic serrated powerlifting bar that has grooves in its material, is 220 cm in total length, has an internal distance between 131 and 132 cm, with a diameter between 2.8 and 2.9 cm, and an external part of 41.5 cm, with a diameter between 5.0/5.2 cm, weighing 20 kg, in addition to 400 kg of washers from Eleiko (Eleiko Sport AB, Halmstad, Sweden).

### 2.3. Blood Pressure

Blood pressure (BP), systolic blood pressure (SBP), diastolic blood pressure (DBP), mean blood pressure (MBP) {MBP = DBP + (SBP − DBP)/3}, and Heart Rate (HR) were measured before and immediately after the training session using a noninvasive automated blood pressure (BP) monitor (Microlife 3AC1-1PC, Microlife, Widnau, Switzerland) [22]. The heart pressure product (HPP) was evaluated according to the following equation: HPP = HR × SBP [2,9]. All BP measurements were taken on the left arm, and the fixation of the cuff on the arm occurred with approximately 2.5 cm of distance between its lower extremity and the antecubital fossae, with the subjects in the sitting position [23,24]. The pre-exercise BP did not exceed 160 and 100 mmHg for SBP and DBP, respectively. Initially, the subjects remained comfortably seated for 10 min in a calm, pleasant environment, and the volunteers were also instructed to avoid the Valsalva maneuver throughout the movement, following guidelines from the American College of Sports Medicine [25].

To assess the occurrence of post-exercise hypotension (PEH), BP and HR were also measured at rest, at exercise peak (immediately following the exercise session), and in the sitting position (at rest) at 5, 10, 20, 30, 40, 50 and 60 min after the exercise session. To obtain MVO_2_, we used a mathematical function based on a high correlation between the product of cardiac pressure and MVO_2_. To estimate myocardial oxygen volume (MVO_2_), a mathematical function was used, expressing the result in ml O_2_/100 g ventilations per minute (VE/min), as follows: MVO_2_ = (HPP × 0.0014) − 6.37 [2,11,26].

### 2.4. Procedures

The training program took place within two weeks with a seven-day interval for each session, and the bench press exercise was used as the only representative of Paralympic weightlifting. The first training session consisted of five sets of 5RM, and all sessions were performed with ≅ 90% of 1RM at week 2, with 5 min of rest between sets [19,27]. Week 1 served for familiarization and the performance of the 1RM test, where each subject started the attempts with a weight they believed they could lift only once using maximum effort, and weight increments were added until reaching the maximum load that could be lifted all at once. If the practitioner could not perform a single repetition, 2.4% to 2.5% of the load used in the test was subtracted [16,28]. Subjects rested for 3–5 min between trials. All subjects underwent two sessions of the 1RM test, with a 48 to 72 h interval between each session.

### 2.5. Statistics

Descriptive statistics were performed using measures of central tendency, mean (X) ± Standard Deviation (SD). The statistical treatment was performed using the computerized package Statistical Package for Social Science (SPSS), version 22.0. To verify the normality of the variables, the Shapiro–Wilk test was used, considering the sample size. To assess the performance between the groups, the ANOVA (two-way) and Bonferroni post hoc tests were performed. The significance level adopted was *p* < 0.05. The effect size was determined by the values of “partial eta squared” (η^2^_p_), considering values of low effect (<0.05), medium effect (0.05–0.25), high effect (0.25–0.50), and very high effect (>0.50) [29,30,31,32]. Statistical analyses were performed using the Statistical Package for Social Science (SPSS) version 25.0 software (IBM, North Castle, New York, NY, USA).

## 3. Results

Our findings are described in Figure 2.

Concerning SBP, there were significant differences between before and after, with an increase in SBP after 50 min between CP and PP (“a” *p* = 0.003), and in PP between the after and 50 min later (“#” *p*= 0.049; η^2^_p_ = 0.576; very high effect). The PP kept the SBP lower after and 60 min later, showing a hypotensive effect present even after 50 min.

In DBP, there were no significant differences between the two groups after 50 min (*p* = 0.38; η^2^_p_ = 0.127; medium effect). In MBP, there were differences between the two groups after 50 min (*p* = 0.012; η^2^_p_ = 0.323; high effect). In BPD and MBP, there were differences in the 50 min later between groups, with lower values for PP.

In the HR, there were no differences between groups. In Group CP, there were differences between before and after (“a” *p* = 0.015), after 5 min (“b” *p* = 0.002), and in relation to 10 min (“c” *p* = 0.003). There were still differences between after 5 min and 40 min (“d” *p* = 0.011), 50 min (“e” *p* = 0.034), and 60 min (*p* = 0.040). In the PP, there were differences between after and 30 min (“g” *p* = 0.021), 40 min (“h” *p* = 0.016), 50 min (“i” *p* = 0.021), and 60 min later (“j” *p* = 0.01; η^2^_p_ = 0.767, very high effect). The PP remained with lower HR values than did the CP.

In HPP and MVO2, there were differences between groups after 60 min (“#” *p* = 0.047). In the CP Group, there were differences between after and 30 min (“a” *p* = 0.012) later, and there were still differences between after 5 min and after 40 min (“b” *p* = 0.006), 50 min (“c” *p* = 0.003), and 60 min (“d” *p* = 0.009). In Group PP, there were differences between after and 50 min later (“e” *p* = 0.023) and 60 min later (“f” *p* = 0.042). There were also differences between after 5 min and after 50 min (“g” *p* = 0.001), and there were differences between after 10 min and after 50 min (“h” *p* = 0.014) and after 60 min (“i” *p* = 0.038; η^2^_p_ = 0.819, very high effect). The PP remained with lower HR values than did the CP.

## 4. Discussion

The goal of our study was to analyze the hemodynamic responses generated by powerlifting exercise in different groups of CP and PP athletes, evaluating the attentive moments and then up to 60 min after the end of the training session. It is noteworthy that the two training methods were performed with high loads (≅90% of 1RM), intended for the training of CP and PP athletes, and the results did not show any risk of cardiovascular overload (HR after session, 5 × 5 method, ~80 bpm). It is worth mentioning that, in this discussion, we use studies of other modalities and with other athletes, given the lack of research focused on this modality and follow-up, with regard to PP. In this direction, corroborating with our findings, another study also worked with intensity (≅90 of 1RM) with five sets of 5RM (5 × 5), in a single powerlifting session; even so, it was not enough to present hemodynamic overload at peak exercise (160 bpm) [11]. Regarding SBP, there were significant differences between before and after, with an increase in SBP after training in both groups (*p* < 0.001), with higher values for CP, although there were no differences between groups. Relating to 60 min, there were significant differences with hypotensive effect for the PP group (*p* = 0.028) in relation to the other moments and to the CP (η^2^_p_ = 0.570, very high effect).

Powerlifting training is characterized by small volumes and high intensities reaching close to 100% or more than 1RM [14,15], being this procedure common to CP and PP. Studies have shown that the magnitude of PHE tends to vary due to changes in training intensity [9,33,34]. Corroborating this, one study found no differences in PHE with variations in training volume [35]. Contrary to this, another study indicated that intensity is not the main factor in PHE but rather volume is [36,37], where high volumes tended to lead to a reduction in BP. There are contradictory results regarding the relationship between volume and intensity in the PHE, which also occurs in resistance exercises [9,33,34]. On the other hand, when evaluating the PHE, comparing resistance training with exercises such as bench press and traditional training, it was reported that PHE was presented 20, 30, and 40 min following training [8].

In BPD, our study did not observe significant differences between the two groups during the evaluation. In MBP, there were differences between before and after (*p* = 0.016, η^2^_p_ = 0.174, mean effect). In HR, there were no differences. Regarding HPP and MVO_2_, there were differences between before and after in both groups (*p* = 0.001, η^2^_p_ = 0.816, very high effect).

The decrease in blood pressure after the strength training indicates that cardiovascular behavior and PHE tend to be influenced by different mechanisms [9,33]. That is, it could be explained by the cardiac output, systolic volume, occlusion of vessels and arteries, autonomic modulation of HR through sympathetic and parasympathetic nerves, in addition to peripheral vascular resistance during training [2,11].

Another study that observed 40% and 80% of 1RM did not notice a hypotensive difference regarding intensities, and BPD tended to decrease at lower intensities [38]. On the other hand, the type of exercise tends to interfere with pressure levels, and intensities of 80% of 1RM tend to influence PHE more in multi-joint exercises, such as bench press, as was used in our study, compared with single-joint exercises, such as fly machine [10]. The hemodynamic responses are associated with the training load (% of 1RM), in addition to the number of sets, repetitions, and density (rest interval between sets), which are important components for PHE [39]. Contrary to this, PHE is associated with the type of exercise, notably involving large muscle groups, and multiarticular work, and not associated with intensity and volume [40].

When there is a comparison between conventional and Paralympic athletes, for conventional athletes, high intensities (>80% 1RM) tend to reduce the cardiac output mediated by systolic volume [41], and in trained individuals, the hemodynamic responses tend to be improved; thus, the decrease in systolic volume compensates for the increase in HR, caused by the increase in sympathetic activity and reduction in parasympathetic activity in the heart [11]. A study with CP athletes, who performed a training session with five sets of 2RM and 5 min of rest between sets, observed PHE 60 min after training, persisting for up to 24 h after exercise [11]. Furthermore, the reduction in pressure levels after resistance exercises is associated with peripheral vasodilators such as nitric oxide, prostaglandins, adenosine, and potassium that tend to influence PHE [42].

In PP, however, there would be a need to increase blood accumulation in the activated region, promoting increased vasodilation and reduced peripheral vascular resistance, providing PHE [40], and considering that training tends to use multiarticular exercises, which normally involve large muscle groups, increasing the PHE. In a study that evaluated the hypotensive effects in Paralympic athletes, no differences were observed between the two training sessions with different intensities, without significant interaction (training vs. time), and there was an increase in SBP immediately after training at both intensities, followed by a decrease to levels below resting values from 20 to 50 min after resistance training [2]. Other studies have presented similar results between conventional and Paralympic athletes in the bench press exercise of powerlifting [28,43,44].

However, our study has some limitations as the mechanisms that promote hypotension were not investigated in the manuscript. We did not investigate peripheral vascularity, sympathetic activity, stroke volume, beta-adrenergic receptors, or endothelial factors, and the method used to assess blood pressure was through validated devices. This has some limitations compared to invasive methods such as intra-arterial catheterization. Thus, every effort was made to ensure that these measurements were obtained in a consistent, reliable, and accurate manner.

## 5. Conclusions

We can conclude that both conventional powerlifting and PP do not present a risk of hemodynamic overload to athletes, considering the HPP results, and that powerlifting tends to promote a moderate hypotensive effect, with an adaptation of blood pressure after and 60 min after exercise. Training through high loads and low volume was shown to be important in terms of hypotensive effect from five minutes to 50 min after training, for CP, and was effective for up to 60 min for PP. Thus, it appears that the hypotensive effect in Paralympic athletes tends to remain longer than in CP.

The findings indicate that strength training for Paralympic athletes tends to be safer, with an increased hypotensive effect and even less post-training increase in BP compared to conventional athletes. Perhaps this is explained by the position of Paralympic athletes, with their lower limbs on the bench and not supported on the floor. Another point to consider is that Paralympic athletes tend to have atrophied lower limbs, which are even more hypotonic. Another point that could contribute to this is the fact that Paralympic athletes are wheelchair users or use crutches to get around, which tends to provide additional adaptation in terms of the use of the upper limbs, which could explain the differences found. However, other studies should be carried out evaluating the hypotensive effect of powerlifting in different physical disabilities and involving females.

## Figures and Tables

**Figure 1 jfmk-06-00092-f001:**
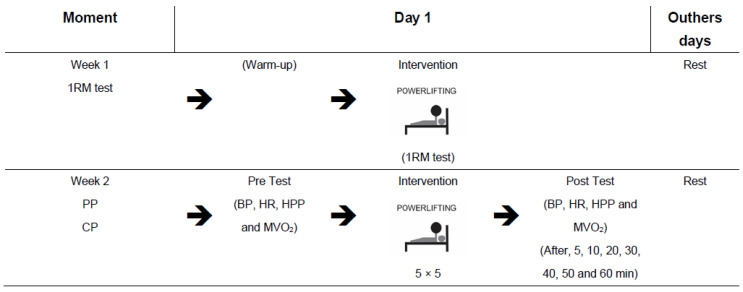
Experimental design—weekly schedule of the tests and intervention. BP: blood pressure, HR: heart rate, HPP: heart pressure product, MVO_2_: myocardial oxygen volume, PP: Paralympic powerlifting. CP: conventional powerlifting, 5 × 5: five series of five maximum repetitions.

**Figure 2 jfmk-06-00092-f002:**
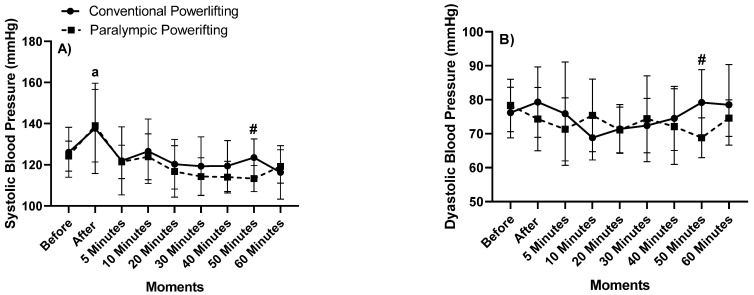
Kinetics of (**A**) systolic blood pressure (SBP), (**B**) diastolic blood pressure (DBP), (**C**) mean blood pressure (MBP), (**D**) heart rate (HR), (**E**) heart pressure product (HPP), and (**F**) MVO_2_. Legend: letters “a–j” indicate intraclass differences, and “#” indicates interclass difference (*p* < 0.05).

**Table 1 jfmk-06-00092-t001:** Characterization of the sample.

	PP X ± SD (CI 95%)	CP X ± SD (CI 95%)	*p*	Cohen’s d	α	ICC
Age (Years)	26.10 ± 6.95 (21.13–31.07)	23.20 ± 2.62 (21.33–25.07)	0.242	0.55	0.208	0.197
Body Mass (Kg)	76.80 ± 17.42 (64.34–89.26)	77.10 ± 7.68 (71.61–82.59)	0.961	0.02	0.419	0.445
Experience (Years)	2.61 ± 0.46 (2.18–2.99)	2.40 ± 0.16 (2.28–2.51)	0.011 *	0.61	0.18	0.017
1RM (Kg)	123.00 ± 29.93 (101.59–144.41)	92.80 ± 9.60 (85.93–99.67)	0.006 *	1.36	0.020	0.011
1RM/Body Mass	1.64 ± 0.39 ** (1.36–1.92)	1.21 ± 0.06 (1.16–1.25)	0.201	1.54	0.049	0.023

* *p* < 0.05. All PP athletes performed with loads that keep them among the 10 bests in their categories at the national level. ** Values above 1.4 in the bench press would be considered elite athletes, according to Ball e Wedman [20]. Legend: PP: Paralympic powerlifting, CP: conventional powerlifting, ICC: intraclass correlation.

## Data Availability

The data that support this study can be obtained from the address: www.ufs.br/DepartmentofPhysicalEducation (accessed on 12 July 2021).

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
