# Peer review of "Evaluation of the Post-Training Hypotensor Effect in Paralympic and Conventional Powerlifting"

_jfmk, 2021, doi:10.3390/jfmk6040092_

Round 1

Reviewer 1 Report

Thank you for the opportunity to review this manuscript comparing PP to CP in the blood pressure response to high load bench press exercise. I have a few considerations for the authors to make regarding their study.

Major comments:

  1. In the introduction, please do not use the word "normal" to describe the CP group - please use "without disability" or a similar phrase to describe this group.
  2. Under the Materials and Methods section first paragraph - the authors need to provide a more complete explanation of the blood pressure measurements. How was blood pressure checked? Were the participants seated or lying? Was an automated cuff used or a sphygmomanometer and stethoscope used? How long had it been since the participants last exercise session when resting BP was measured? Was BP checked more than once and averaged or only once with each measurement?
  3. How was "without disabilities" defined in this study?
  4. How was a recreationally active individual defined in this study? Is this the most appropriate population to compare to with the high-performance PP athletes?

Minor comments:

5. In the abstract, I believe the authors are referring to "mean blood pressure" not "medium blood pressure"?

6. The first sentence of the introduction is awkward and incomplete - please amend.

7. Line 78 - do you mean 5RM instead of 1RM?

8. Line 136 - should be 20 kg not 20 lg bar.

9. Lines 155 and 208 - should be DBP as abbreviation.

10. Line 228 - use the word "exercise" instead of "training".

Author Response

Reviewer 1

Thank you for the opportunity to review this manuscript comparing PP to CP in the blood pressure response to high load bench press exercise. I have a few considerations for the authors to make regarding their study.

Major comments:

  1. In the introduction, please do not use the word "normal" to describe the CP group - please use "without disability" or a similar phrase to describe this group.

Answer: Adjusted, line 70.

  1. Under the Materials and Methods section first paragraph - the authors need to provide a more complete explanation of the blood pressure measurements. How was blood pressure checked? Were the participants seated or lying? Was an automated cuff used or a sphygmomanometer and stethoscope used? How long had it been since the participants last exercise session when resting BP was measured? Was BP checked more than once and averaged or only once with each measurement?

Answer: Adjusted. It is described in detail in lines 148 to 166.

  1. How was "without disabilities" defined in this study?

Answer: Adjusted, line 107.

  1. How was a recreationally active individual defined in this study? Is this the most appropriate population to compare to with the high-performance PP athletes?

Answer: Adjusted, lines 105-107.

Minor comments:

  1. In the abstract, I believe the authors are referring to "mean blood pressure" not "medium blood pressure"?

Answer: Adjusted, line 40.

  1. The first sentence of the introduction is awkward and incomplete - please amend.

Answer: Adjusted, Lines 54-57.

  1. Line 78 - do you mean 5RM instead of 1RM?

Answer: Adjusted.

  1. Line 136 - should be 20 kg not 20 lg bar.

Answer: Adjusted.

  1. Lines 155 and 208 - should be DBP as abbreviation.

Answer: Adjusted. However, we have already corrected for heart pressure product (HPP).

  1. Line 228 - use the word "exercise" instead of "training".

Answer: Adjusted.

Reviewer 2 Report

Overall this is very interesting study. I have several important comments/suggestions

  1. What "#" means in the Figure 2 needs to explain in the figure legend. Also, "a, b, c, d, e, f..." in the figure 2 needs to describe in the figure legend
  2. What is the future direction and clinical implication of this study, need additional discussion prior to conclusion
  3. Regarding "Participated in the study 10 CP athletes, who trained recreationally, without disabilities, and 10 male PP athletes, participating in the project at the Federal University of Sergipe". All 10 CP are males? Or any females? given 10 PP are males, is this fair comparison. In addition why only males and no females? This may lead to gender bias.

Author Response

Reviewer 2

Overall this is very interesting study. I have several important comments/suggestions

  1. What "#" means in the Figure 2 needs to explain in the figure legend.

Answer: Adjusted, Line 202.

Also, "a, b, c, d, e, f..." in the figure 2 needs to describe in the figure legend

Answer: Adjusted, Line 202.

  1. What is the future direction and clinical implication of this study, need additional discussion prior to conclusion

Answer: Adjusted, Lines 320, 321

  1. Regarding "Participated in the study 10 CP athletes, who trained recreationally, without disabilities, and 10 male PP athletes, participating in the project at the Federal University of Sergipe".

Answer: Adjusted, Line 108.

All 10 CP are males? Or any females? given 10 PP are males, is this fair comparison. In addition why only males and no females?

Answer: The sample was intentional, and thus, we do not have enough females to make a comparison, both in the CP and PP. Perhaps this is explained considering that the number of male athletes is much greater than the number of female athletes in this modality.

This may lead to gender bias.

Answer: As mentioned, it is not about prejudice, a smaller number of female practitioners.

Round 2

Reviewer 1 Report

Thank you for addressing my queries and comments. Just two minor corrections need to occur to the current version of the manuscript.

  1. In the abstract high blood pressure is abbreviated AH and in the introduction is abbreviated HBP - please be consistent with the abbreviation chosen.
  2. Line 158 uses BPD as the abbreviation for diastolic blood pressure when it should be DBP - please amend.

Author Response

all corrections are done as requested